# Image Denoising Based on GAN with Optimization Algorithm

**Min-Ling Zhu [1], Liang-Liang Zhao [1] and Li Xiao [2,3,*]**

1 Computer School, Beijing Information Science and Technology University, Beijing 100101, China
2 Key Laboratory of Intelligent Information Processing, Institute of Computing Technology Chinese Academy of Sciences, Beijing 100090, China
3 Ningbo Huamei Hospital, University of Chinese Academy of Sciences, Ningbo 315010, China
* Correspondence: andrew.lxiao@gmail.com

**Abstract:** Image denoising has been a knotty issue in the computer vision field, although the developing deep learning technology has brought remarkable improvements in image denoising. Denoising networks based on deep learning technology still face some problems, such as in their accuracy and robustness. This paper constructs a robust denoising network based on a generative adversarial network (GAN). Since the neural network has the phenomena of gradient dispersion and feature disappearance, the global residual is added to the autoencoder in the generator network, to extract and learn the features of the input image, so as to ensure the stability of the network. On this basis, we proposed an optimization algorithm (OA), to train and optimize the mean and variance of noise on each node of the generator. Then the robustness of the denoising network was improved through back propagation. Experimental results showed that the model's denoising effect is remarkable. The accuracy of the proposed model was over 99% in the MNIST data set and over 90% in the CIFAR10 data set. The peak signal to noise ratio (PSNR) and structural similarity (SSIM) values of the proposed model were better than the state-of-the-art models in the BDS500 data set. Moreover, an anti-interference test of the model showed that the defense capacities of both the fast gradient sign method (FGSM) and project gradient descent (PGD) attacks were significantly improved, with PSNR and SSIM values decreased by less than 2%.

**Keywords:** image denoising; GAN; optimization algorithm; autoencoder; ResNet





## 1. Introduction

Image denoising is one of the hottest research topics in the field of image processing [1]. There are various traditional image denoising methods. Tang used an improved curvature filtering algorithm, where a projection operator was used to replace the minimum triangular tangent plane projection operator of the traditional curvature filtering [2]. Li proposed an adaptive matching and tracking algorithm. First, the sparse coefficients were calculated. Then the dictionary was trained to be an adaptive dictionary, which could reflect the image structure effectively by using the K singular value decomposition algorithm. Finally, the image was reconstructed by combining the sparse coefficients with the adaptive dictionary [3]. Dabov proposed block-matching and 3D filtering (BM3D), which made use of the self-similarity existing in natural images to match with adjacent image blocks, and then the similar blocks were integrated to form the denoised image through domain transformation [4]. Xu proposed a trilateral weighted sparse coding (TWSC) scheme for robust real image denoising [5]. Xie proposed a non-convex regular low rank sparse matrix decomposition method for image denoising [6]. Although the above traditional denoising methods achieved a good effect to a certain degree, there are highly time consuming and low robustness. Li proposed a new image denoising approach based on undecimated discrete wavelet transform (UDWT), which combines the technique of cone of influence (COI) analyzing and UDWT [7].

In recent years, with the rapid development of deep learning and remarkable achievements in the field of image processing, more and more people are applying deep learning to image denoising. For example, the convolutional neural network has two major characteristics, of local perception and parameter sharing, which have a good effect in image feature extraction and recognition. Wang proposed a gradient vector convolution (GVC) model for image denoising [8]. Wu proposed an interleaved cascade of shrinkage fields (CSF) to reduce noise and jointly restore the transmission diagram and scene radiance from a single noise image [9]. Zhang proposed a feedforward denoising convolutional neural network (DnCNN) model, which combined batch normalization and residual learning [10]. Yan proposed a self-consistent GAN network (SCGAN) to extract noise images directly from noisy images, to achieve unsupervised noise modeling [11]. Yu proposed a deep iterative down-up convolutional neural network (DIDN) for image denoising, which can process various noise levels using a single model, without input noise information as a solution [12]. Zhang proposed a fast and flexible denoising convolutional neural network (FFDNet), which used a noise estimation graph as input, balancing the suppression of uniform noise and the preservation of details [13]. Chen's proposed denoising method used GAN to model the noise distribution, to generate noise samples through the established model and form a training data set with clean image sets, and to train the denoising network model to perform blind denoising [14]. Dong proposed a convolutional neural network denoising method based on multi-scale redundancy of natural images [15]. Wang proposed a novel channel and spatial attention neural network for image denoising [16]. Cai proposed a new efficient image denoising scheme, where global structure and local similarity preservations combined method of optimal directions (MOD) with approximate K-SVD (AK-SVD) for dictionary learning [17]. Cai proposed a new development of non-local image denoising using fixed-point iteration for non-convex $\ell$p sparse optimization [18]. Although neural networks are widely applied in the field of image processing, they are vulnerable to adversarial attacks that lead to incorrect network outputs. In 2014, Szegedy Christian introduced the L-BFGS method, which induced the model to obtain a result completely deviating from the real value by adding slight disturbance to the input sample image of the model [19]. In 2015, Goodfellow Ian J proposed an adversarial sample generation algorithm based on the fast gradient sign method (FGSM), which sought the direction with the largest gradient change in the deep learning model and generated disturbances, to increase the loss of image classifiers in this direction [20]. Later, the FGSM derived project gradient descent (PGD) and other gradient-based attack algorithms. However, some current defense methods require a lot of manpower and material resources and have poor robustness [21].

In view of low robustness of traditional denoising methods and vulnerability of deep learning network under attacks, this paper introduces a simple and efficient method to improve the robustness of the denoising network. The whole backbone of the denoising-network is based on the GAN. Moreover, the denoised image is from the GAN. Random noise is added into the neural network and it is optimized through back propagation. The most important feature is that this method does not require additional resource consumption and can simultaneously improve the model's ability for denoising and defense against attack. Furthermore, an integrated image denoising network is designed. Finally, FGSM and PGD attack experiments were used to verify the anti-interference capability of the adversarial network.

## 2. Related Work

In this section, we briefly overview some of the basic network modules and loss functions that are involved in our design. First, we refer to the following three networks: The first is the autoencoder, which is a form of neural network and is composed of an encoder and decoder [22]. The encoder compresses the original data to obtain the features of the original data, and learns the features through other neural networks to reduce the burden of network generation. The decoder decompresses the learned features into original data. This is an unsupervised algorithm, and then the back propagation algorithm is used

to train the network to make the output close to the standard image. The second is the residual module [23]. Although more features can be extracted, the training is also more difficult due to the increasing depth of the neural network. With the increase of depth, the original data information will be gradually lost in the process of convolution and pooling, and the error signal is prone to gradient dispersion during the back propagation. Therefore, the residual network is introduced to solve the training difficulties caused by increasing the network depth. The residual network uses jump connections to connect the features after convolution and pooling with the previous features, and the information representation is enhanced by the addition of both gradual and deep features. This method avoids the problem of image feature loss due to the increase of network depth, and solves the problem of gradient dispersion and ensures the stability of the network. The third aspect is the generative and adversarial network based on the two-person game idea, which is widely used in various aspects of the imaging field. A generative adversarial network is a method of unsupervised learning. It consists of a generator network and a discriminator network, and learns by playing two neural networks against each other. The generator network takes random samples from the latent space as input, and its output should imitate the real samples in the training set as much as possible. The input of the discriminator network is the real sample or the output of the generator network, and the purpose of the discriminator network is to distinguish the output of the generator network from the real sample as far as possible. The generator network tries to deceive the discriminator network as much as possible. The final purpose of the two networks is to make the discriminator network unable to judge whether the output result of the generator network is true or not [24].

Furthermore, we refer to three loss functions. The first is MSE loss [25]. The values of each pixel of the generated image and the original image are compared, and the mean square error of the generator network is represented by the loss of pixels. The second is GAN loss, which is mainly formed by the discrimination network to determine between the generated denoised image or the original real image [26]. The GAN loss ensures that the generator network generates an image as close to the real image as possible. Then the discriminator network is deceived, to achieve the optimal result of the generated image. The third is classification loss [27]. As the generated image may cause the loss of some features, it is necessary to analyze the generated image category. Then the generator network can generate the same image as the real image, as far as possible.

## 3. Network Structure Design and Optimization Algorithm

The whole network structure is based on GAN. The generator network uses an autoencoder for image generation. A discriminator network is used to discriminate between the generated images. When the discriminator network cannot discriminate the authenticity of the generated images, the generated images can be used as the input of a classification network, to further verify the denoising ability of the network for noisy images. On the other hand, Gaussian noise is added to the stochastic gradient estimates of the standard deviation path of each neural network neuron. In this way, the gradient estimates and the noise level are byproducts of back propagation.

### 3.1. Whole Network Structure Design

The network framework we proposed is shown in Figure 1. It consists of three subnetworks: a generator network (G), discriminator network (D), and classification network (C). The G inputs an image with noise and outputs an image with the same size as the original image, through feature extraction of the network; the D inputs the generated image and standard image, and outputs "0" or "1", which represent the similarity between the generated image and standard image; the C inputs generated images, to complete the classification of image content. In G and D, we apply the network optimization algorithm (OA) proposed in the following section, which improves the robustness of GAN networks. The MSE loss and GAN loss are used to update the iterative training parameters of the

GAN neural network; classification loss is used to update the iterative training parameters of the classification network. The training finally makes the network tend to be stable.

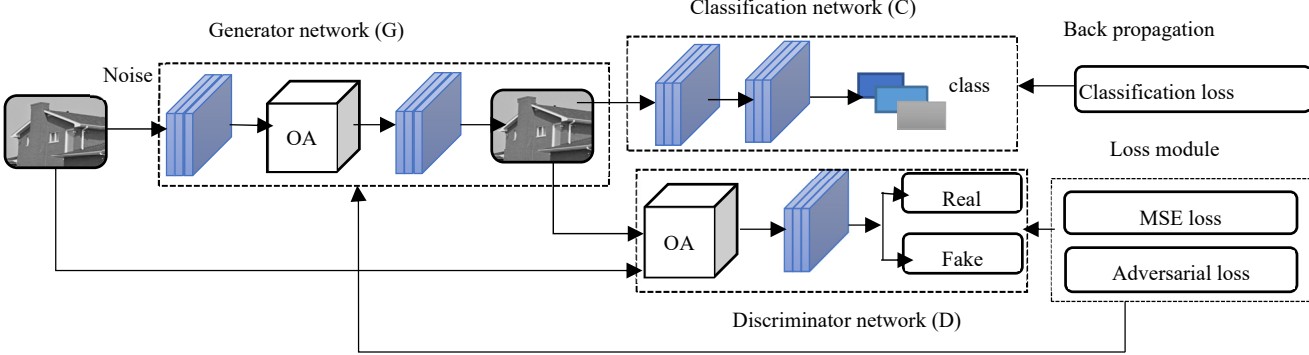

**Figure 1.** Whole network structure.

### 3.2. Optimization Algorithm

Here we deduce the OA in Figure 1. Let $\tau$ represent the layers of the neural network; $m_t$ represents the number of neurons at layer $t$, $t \in 1, 2, \ldots, \tau$. The output of layer $t$ is $x^{(t)} = [x_1^{(t)}, x_2^{(t)}, \ldots, x_{m_t}^{(t)}] \in \mathrm{R}^{m_t}$, and $x^{(0)}$ is the input of the network.

Suppose the network has $N$ inputs, denoted as $x^{(0)}(N)$, $N = 1, 2, \ldots, n$. For the $n$ input, the $i$ output of the $t$ layer is Formulas (1) and (2).

$$x_i^{(t+1)}(n) = \varphi\left(v_i^{(t)}\right) \tag{1}$$

$$v_i^{(t)} = \sum_{j=0}^{m_t} \theta_{i,j}^{(t)} x_j^{(t)}(n) + z_i^{(t)}(n) \tag{2}$$

$x_j^{(t)}(n)$ is the $j$ input of the $n$ data in the $t$ layer; $\theta_{i,j}^{(t)}$ is the weight of the $i$ input in the $t$ layer; $v_i^{(t)}$ is the $i$ output of the $t$ layer; $\varphi$ is the activation function; $z_i^{(t)}(n)$ is the $n$ data and independent random noise added to the $i$ neuron in the $t$ layer. Figure 2 shows a visualization of noise addition.

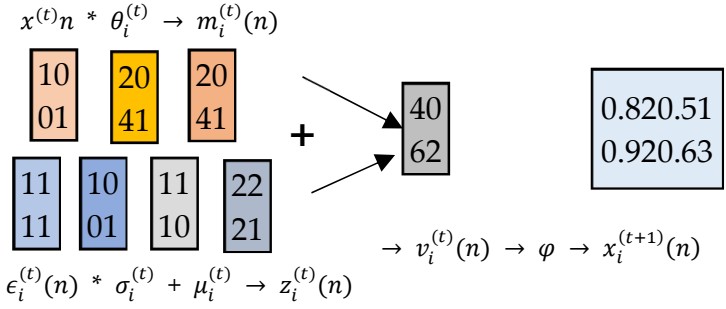

**Figure 2.** Optimization algorithm visualization.

$L$ represents the loss function. For the $n$ data $x^{(0)}(n)$ marked as $Y(n)$, $L(x^{(\tau)}(n), Y(n))$ represents the loss value. In our work, we tried to optimize the size of the noise level of the central normal random noise $\sigma_i^{(t)}$ of each neuron. $z_i^{(t)}(n) = \sigma_i^{(t)} \varepsilon_i^{(t)}(n)$, where $\varepsilon_i^{(t)}(n)$ is a

standard normal random variable. The residual of the *i* neuron at the *t* layer of the *n* data propagates backward through the neural network and is defined as as Formula (3).

$$
\delta_i^{(t)}(n) = \begin{cases} e_i^{(\tau)}(n)\varphi'\left(v_i^{(\tau-1)}(n)\right) & t = \tau \\ \varphi'\left(v_i^{(t-1)}(n)\right)\left(\sum_{j=0}^{m_k}\theta_{i,j}^{(t)}\delta_j^{(t+1)}(n)\right) & t < \tau \end{cases}
\tag{3}
$$

$e_i^{(\tau)}(n)$ is defined as formula (4):

$$
e_i^{(\tau)}(n) = \left.\frac{\partial L(x, Y(n))}{\partial x_i}\right|_{x=x^{(\tau)}(n)}
\tag{4}
$$

Back propagation essentially provides information about all parameters $\theta_{i,j}^{(t)}$ ($t=1,2,\ldots\tau-1$), path random derivative estimation of loss function *L*. As shown in Formula (5), $j \in \{0, 1,\ldots, m_t\}$, $i \in \{0, 1,\ldots, m_{t+1}\}$.

$$
\frac{\partial L\left(x^{(\tau)}(n), Y(n)\right)}{\partial\theta_{i,j}^{(t)}} = \delta_j^{(t+1)}(n)x_j^{(t)}(n)
\tag{5}
$$

The algorithm flow is as follows:

(a)    First input training data $P = \left\{\left(x^{(0)}(n),\ Y(n)\right)\right\}_{n=1}^{N}$, loss function *L*.

(b)    Construct neural network.

(c)    Use Formulas (1) and (2) to calculate the output $x^{(\tau)}(n)$.

(d)    Calculate the loss function $L\left(x^{(\tau)}(n),\ Y(n)\right)$.

(e)    Use Formulas (3) and (5), respectively, to estimate the gradient of loss to weight and noise level.

(f)    Update weights and noise levels.

(g)    Repeat steps c to f until the parameters meet the requirements of the model.

### 3.3. Sub-Network Structure Design

The three sub-network structures proposed in this paper are shown in Figure 3.

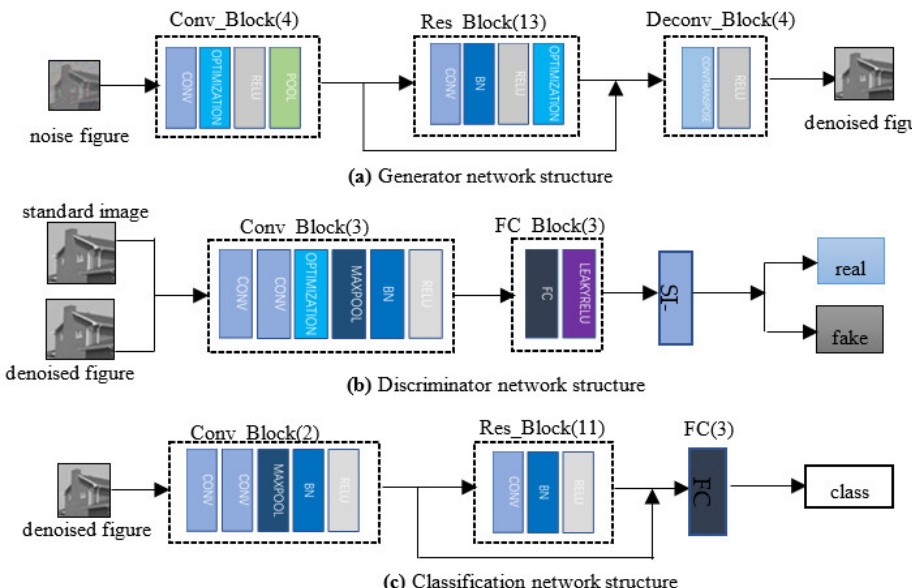

**Figure 3.** Sub-network structures.

Figure 3a shows the network structure of the generator network, which includes four convolution blocks, thirteen residual blocks, and four deconvolution blocks. Each one of four convolution blocks includes a convolution layer, optimization layer, relu layer, and pooling layer. In addition, each of thirteen residual blocks includes a convolutional layer, batch normalization layer, relu layer, and algorithm optimization layer. While, each one of the four deconvolution blocks includes a deconvolution layer and relu layer. The network outputs an image the same size as the standard image. The generator network is the core part of the whole network, and the image denoising effect largely depends on the ability of the generator network. Therefore, the neural network adopts encoding and decoding structures such as the autoencoder. A residual module jump connection is added in the middle, to enhance image feature representation, to avoid gradient dispersion, and to ensure the stability of the network.

Figure 3b shows the network structure of the discriminator network, which includes three convolution blocks, three linking blocks, and a sigmoid function layer. Each of three convolution blocks includes two convolution layers, an optimization layer, maximum pooling layer, batch normalization layer, and relu layer. Each of the three linking blocks includes a full link layer and leakyrelu layer. The sigmoid function layer outputs "0" or "1", which is used for the binary classification problem, to judge the difference between the positive and negative labels of the image. The discriminator network is designed based on the full convolution neural network, to discriminate the similarity between the standard image and the generated image.

Figure 3c shows the network structure of the classification network, which includes two convolution blocks, eleven residual blocks, and three full connection layers. Every two convolution blocks include a maximum pooling layer, batch normalization layer, and relu layer. Each of the eleven residual blocks includes a convolution layer, batch normalization layer, and relu layer. The final full connection layer outputs n categories to complete the classification of images. The classification network is used to classify the generated-images after the optimization of the generated network.

## 4. Experiments and Analyses

First, the proposed method was used to test the classification accuracy in the MNIST and CIFAR10 data sets. Then the method was compared with the DnCNN, BM3D, FFDNet, and IRCNN denoising methods, and the PSNR and SSIM values were calculated, which under the standard deviation of Gaussian noise were 25, 50, 75, and 100. Moreover, we performed a visual perception experiment. Finally, the network robustness was verified under FGSM and PGD attacks. The experiments illustrated that the method is effective.

### 4.1. Data Set and Parameter Setting

The MNIST data set is very well known. It consists of 60,000 training samples and 10,000 test samples, where each sample is a $28 \times 28$ pixel grayscale handwritten digital image. The Cifar-10 data set contains 50,000 training images and 10,000 test images, all of which are 3-channel color RGB images with a size of $32 \times 32$, including 10 categories in total. The two data sets were used to test the accuracy of model recognition under different noise conditions. Then we used the BDS500 data set to train and test the model. The peak signal to noise ratio (PSNR) and structural similarity (SSIM) were compared with other methods under different noise conditions.

The hardware platform of this experiment was a Tesla P100 with 16GB memory; software was Ubuntu18.04, CUDA10.02, python3.6; and the deep learning framework was Pytorch1.8; the batch processing was 128; the Adam algorithm was used to update the gradient; the initial learning rate was 0.001, and the learning rate decreased as the number of trainings increased; the momentum was 0.9.

### 4.2. Evaluation Index

The fidelity of image denoising is represented by the evaluation index, which is the error between the standard image and the denoised image, and the PSNR and SSIM are used for evaluation and analysis.

PSNR measures denoising performance, using the error between corresponding pixels of the denoising image and the standard image. PSNR is expressed as Formulas (6) and (7).

$$MSE = \frac{1}{mn} \sum_{i=0}^{m-1} \sum_{j=0}^{n-1} [I(i,j) - K(i,j)]^2 \tag{6}$$

$$PSNR = 10\lg\frac{MAX_I^2}{MSE} \tag{7}$$

where $m$ and $n$ represent the number of rows and columns of the image pixels, $MAX_I$ is the maximum possible pixel value of the image. According to Formulas (6) and (7), the larger $MSE$ is, the smaller $PSNR$ is, which indicates that the denoising effect is good and the denoised image is closer to the standard image.

SSIM is measured based on the luminance, contrast, and structure between the denoised image and standard image. The value ranges from "0" to "1", a larger value indicates a better denoising effect. SSIM is expressed as Formulas (8) and (9).

$$\begin{cases} l(x,y) = \dfrac{2\mu_x\mu_y + c_1}{\mu_x^2 + \mu_y^2 + c_1} \\[2mm] c(x,y) = \dfrac{2\sigma_x\sigma_y + c_2}{\sigma_x^2 + \sigma_y^2 + c_2} \\[2mm] s(x,y) = \dfrac{\sigma_{xy} + c_3}{\sigma_x\sigma_y + c_3} \end{cases} \tag{8}$$

$$SSIM(x,y) = \frac{(2\mu_x\mu_y + c_1)(2\sigma_{xy} + c_2)}{\left(\mu_x^2 + \mu_y^2 + c_1\right)\left(\sigma_x^2 + \sigma_y^2 + c_2\right)} \tag{9}$$

$\mu_x$ is the mean value of $x$; $\mu_y$ is the mean value of $y$; $\sigma_x^2$ is the variance of $x$; $\sigma_y^2$ is the variance of $y$; $\sigma_{xy}$ is the covariance of $x$ and $y$; $c_1 = (K_1L)^2, c_2 = (K_2L)^2$ which are constants that avoid zero; $L$ is the range of pixel value; $K_1$=0.01 and $K_2 = 0.03$ are the default values.

### 4.3. Experimental Result and Analysis

4.3.1. Comparison of Classification Accuracy on Different Data Sets

In this paper, Gaussian noises with standard deviations of 25, 50, and 75 were added to the test set. The experimental results are shown in Figure 4.

From Figure 4a, we can see that under the influence of different noise environments the classification accuracy could reach more than 99%, and the experimental error remained within 0.005. This proves that the method is feasible for image denoising. It can resolve the classification problem of different noise levels and the images can be correctly classified under different noise levels.

Figure 4b shows the classification accuracy on CIFAR10, which could reach more than 90%. CIFAR10 is a rebuilt data set including RGB images with noise, so that the classification of CIFAR10 was harder. The experimental results showed the experimental error was stable within ±0.1. This shows that the algorithm not only had a significant denoising effect for grayscale images, but also had a strong denoising ability for RGB color images, and it could realize the classification of color images and ensure the recognition accuracy of images. This paper mainly compared the accuracy gap between denoised images and standard images, without excessively pursuing the recognition accuracy of the

data set. Therefore, the recognition of the data set did not achieved an optimal effect, which will be the next project.

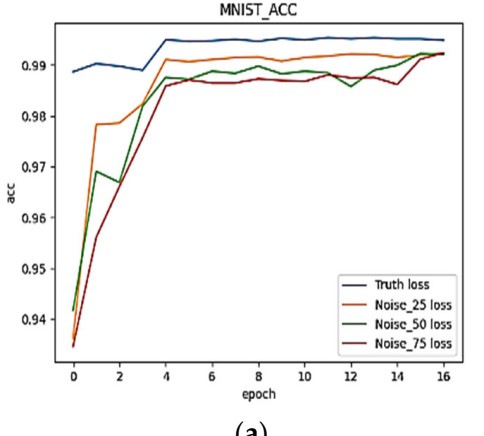
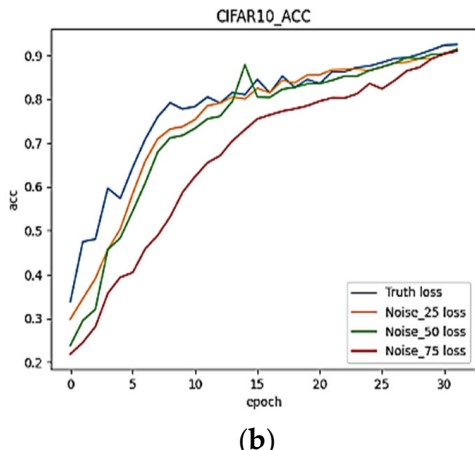

(**a**)　　　　　　　　　　　　　　　　　　　(**b**)

**Figure 4.** MNIST and CIFAR10 classification accuracy. (**a**) classification accuracy on the MNIST data set, (**b**) classification accuracy on the CIFAR10 data set.

4.3.2. Comparison of PSNR and SSIM on the BDS500 Data Set among Different Methods

To compare the PSNR and SSIM values after denoising, Gaussian noises with standard deviations of 25, 50, 75, and 100 were added to the images from the BDS500 data set. Then the DnCNN, BM3D, FFDNet, IRCNN, LSLA-2, UDWT, and our method were tested. The results are shown in Tables 1 and 2.

**Table 1.** PSNR values of the different methods.

| Noise ($\sigma$) | BM3D | UDWT | DnCNN | FFDNet | IRCNN | LSLA-2 | This Paper |
|---|---|---|---|---|---|---|---|
| 25 | 29.97 | 25.51 | 30.43 | **30.44** | 30.38 | 28.99 | 27.53 |
| 50 | 26.72 | 23.42 | 27.18 | **27.32** | 26.32 | 25.63 | 26.85 |
| 75 | 22.32 | 19.98 | 22.21 | 22.43 | 22.87 | 22.31 | **24.49** |
| 100 | 19.56 | 17.53 | 20.12 | 20.62 | 19.78 | 20.54 | **24.71** |

**Table 2.** SSIM values of the different methods.

| Noise ($\sigma$) | BM3D | UDWT | DnCNN | FFDNet | IRCNN | LSLA-2 | This Paper |
|---|---|---|---|---|---|---|---|
| 25 | 0.8447 | 0.8053 | **0.8597** | 0.8582 | 0.8576 | 0.8286 | 0.8413 |
| 50 | 0.7659 | 0.7495 | 0.7865 | 0.7841 | 0.7853 | 0.7664 | **0.8176** |
| 75 | 0.7132 | 0.7054 | 0.7178 | 0.7232 | 0.7152 | 0.7143 | **0.7868** |
| 100 | 0.6856 | 0.6394 | 0.6871 | 0.6882 | 0.6725 | 0.6532 | **0.7640** |

It can be seen from Table 1 that the PSNR values of BM3D, DnCNN, FFDNet, IRCNN, UDWT, and LSLA-2 are slightly higher than this paper's method, when the standard deviation of Gaussian noise $\sigma = 25$, and the difference was almost the same when the standard deviation of Gaussian noise $\sigma = 50$, even being slightly higher than that of some methods. When the standard deviation of Gaussian noise was $\sigma > 50$, the proposed method was significantly higher than the other methods. When the standard deviation of Gaussian noise $\sigma > 50$, the PSNR of the proposed method was about 4 dB higher than the other methods.

Table 2 shows that the SSIM value of the proposed method was lower than that of other methods when $\sigma = 25$; and the SSIM value of the proposed method was significantly higher than that of the other methods when standard deviation of Gaussian noise was greater than 25.

### 4.3.3. Comparison of Visual Perception

In view of the evaluation index of visual perception difference, this paper selected a picture in the test set for visualization under different methods. The experimental results are shown in Figure 5. Where (a) is the standard image; (b) is the image with Gaussian noise; (d) is the image denoised by BM3D; © is the image denoised by DnCNN; (f) is the image denoised by FFDNet; and (g) is the image denoised by IRCNN. Although these methods also removed the noise of the image, the image looks partly fuzzy and some edge features have a fuzzy phenomenon. The image (c), denoised by the method proposed in this paper, has a more intuitive visual experience. The clarity of the denoised image is almost the same as that of the standard image, and the features of the image are relatively intact. The image in this paper is clearer.

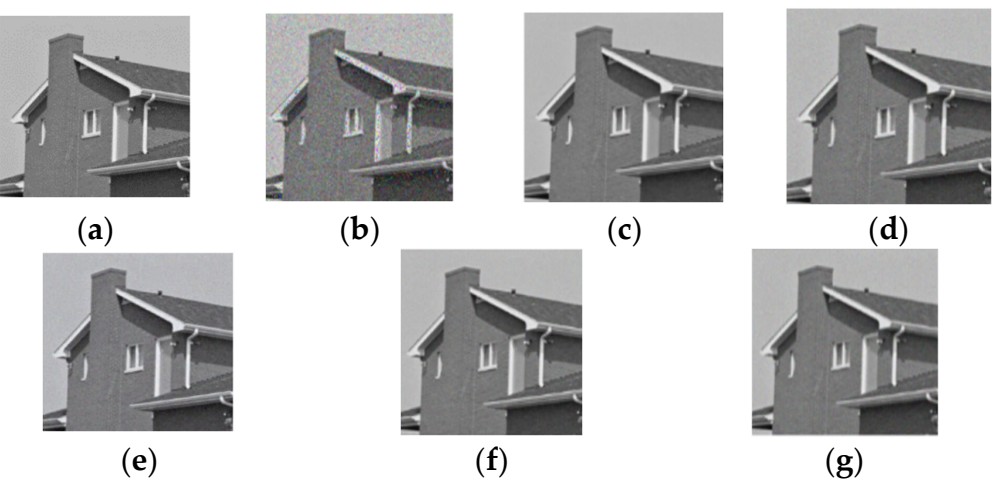

**Figure 5.** Image denoised using different methods. (**a**) original image, (**b**) noise image, (**c**) this paper, (**d**) MB3D, (**e**) DnCNN, (**f**) FFDNet, (**g**) IRCNN.

To sum up, when the noise level was low, the denoising effect of the method in this paper was equal to that of the other methods. However, when the noise standard deviation was greater than 25, the denoising ability and effect of the proposed method were better than the other methods, and both the values of PSNR and SSIM were higher than other methods. The test showed that when the noise environment was more complex, our method was more advantageous and had a stronger robustness and could effectively improve the image. This paper's method had little influence on the noise environment but its denoising ability was relatively stable in different environments.

### 4.3.4. FGSM Attack Result

FGSM is an algorithm based on gradient generation of adversarial samples and is a single-step, non-directional attack algorithm. Figures 6 and 7 show the comparison effect of SSIM and PSNR values between the generated images and the standard images under different attack degrees. The range of difference between the SSIM and PSNR values of the generated image and the standard image become smaller with a larger disturbance after FGSM attacks. Therefore, the method of adding random noise to the neurons of a neural network can improve the anti-interference ability of the network, which proved the superiority of our method in stability and robustness.

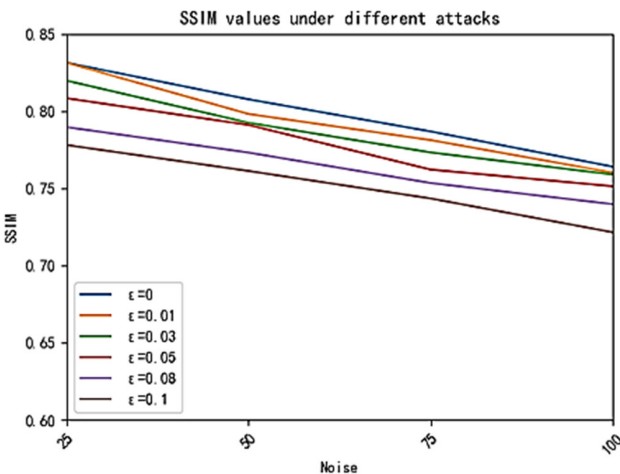

**Figure 6.** SSIM values under different levels of FGSM attacks.

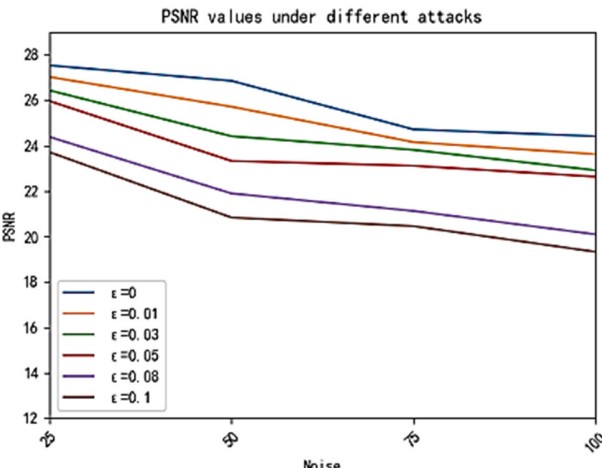

**Figure 7.** PSNR values under different levels of FGSM attacks.

4.3.5. Ablation Experiments and PGD Attack

In order to further verify the restoration ability of this paper's method with noisy images, an ablation experiment was carried out. First, the optimization algorithm (OA) was removed, to test the performance of the model. Gaussian noise with a standard deviation of 25, 50, 75, and 100 was added to the BDS500 dataset for the experiment. Comparing the PSNR and SSIM, the results are shown in Table 3. When OA was used in the generator network and discriminator network, it could optimize the network and achieve better results in the processing of noise images. This shows that our optimization method could improve the robustness of the network.

**Table 3.** Results of ablation experiments with no PGD (PSNR/SSIM).

|  | $\sigma$=25 | $\sigma$=50 | $\sigma$=75 | $\sigma$=100 |
| --- | --- | --- | --- | --- |
| **With OA (PSNR/SSIM)** | 27.53/0.8413 | 26.86/0.8176 | 24.49/0.7868 | 24.71/0.7640 |
| **Without OA (PSNR/SSIM)** | 21.13/0.6396 | 20.45/0.6034 | 19.12/0.5958 | 18.63/0.5756 |

Second, in order to further verify the robustness of this paper's method for the network, experiments with OA and without OA were performed, to test the defense performance of the model under different disturbance levels of PGD adversarial attack. The PGD attack is an iterative attack, which can be regarded as a copy of FGSM–K-FGSM (K represents the number of iterations). We performed a 10-step PGD adversarial training with a step size of

0.01, to verify the stability of the model under different disturbance levels. The results are shown in Table 4. The defense performance of the network against PGD attack decreased significantly without OA. With the increase of attack amplitude, the SSIM and PSNR values without OA decreased more than those of the network with OA. When $\epsilon = 0.05$, adding OA could even improve the SSIM and PSNR values by more than 100%. This proved that adding OA could improve the anti-interference ability and enhance the robustness of the network.

**Table 4.** Results of ablation experiments under PGD (PSNR/SSIM).

| | | $\sigma$=25 | $\sigma$=50 | $\sigma$=75 | $\sigma$=100 |
|---|---|---|---|---|---|
| **With OA (PSNR/SSIM)** | $\epsilon = 0.01$ | 26.93/0.8325 | 25.86/0.8123 | 23.91/0.7783 | 24.02/0.7601 |
| | $\epsilon = 0.02$ | 26.52/0.8297 | 25.21/0.8043 | 23.42/0.7642 | 23.02/0.7554 |
| | $\epsilon = 0.05$ | 26.36/0.8223 | 25.15/0.7931 | 22.97/0.7662 | 22.25/0.7510 |
| **Without OA (PSNR/SSIM)** | $\epsilon = 0.01$ | 16.57/0.5217 | 15.50/0.5020 | 14.35/0.4715 | 13.36/0.4563 |
| | $\epsilon = 0.02$ | 13.45/0.4570 | 12.62/0.4234 | 11.98/0.4044 | 10.52/0.3851 |
| | $\epsilon = 0.05$ | 11.39/0.4178 | 10.84/0.3899 | 10.02/0.3620 | 9.15/0.3572 |

## 5. Conclusions

This paper proposed an image denoising method based on GAN network. In our method, a global residual is added into the autoencoder to extract and learn the features of the input image, preventing the loss of features in the process of denoising and preserving the details of the image features. Gaussian noise is added to the standard deviation path random estimation of each neuron in the neural network, to make it become a by-product of back propagation, which can effectively increase the robustness of the neural network and make it relatively stable in the case of noise environment fluctuations. MSE loss and adversarial loss are used to adjust the network, so that the network can achieve the best performance and have a better denoising effect. We compared our method with other methods. Although it was not as good as the other methods in the case of a low noise level, it was generally better than the other methods in the case of a high noise level. Both from the perspective of vision and quantitative objective evaluation, the denoising effect of the proposed method was remarkable in most scenes. The algorithm model provides help for target detection, recognition, and other applications, and it also has a good practicability. The future work after this paper is to further optimize the denoising effect in low noise environments, so as to achieve an optimal denoising effect in all noise environments

**Author Contributions:** Conceptualization: M.-L.Z. and L.X., methodology: M.-L.Z. and L.X., formal analysis: M.-L.Z. and L.-L.Z., investigation: M.-L.Z. and L.X., data curation: M.-L.Z. and L.-L.Z., writing– original draft preparation: M.-L.Z. and L.-L.Z., writing–review and editing: M.-L.Z. and L.-L.Z. All authors have read and agreed to the published version of the manuscript.

**Funding:** This research was funded by the Beijing Natural Science Foundation (No. 4202025), National Natural Science Foundation of China (No. 31900979) and Promoting the classified development of colleges and universities—the construction of the first level discipline of Computer Science and Technology (No. 5112211036).

**Institutional Review Board Statement:** Not applicable.

**Informed Consent Statement:** Not applicable.

**Data Availability Statement:** Not applicable.

**Acknowledgments:** We thank the company ZSE, a.s., for supporting the open-access publication of this paper.

**Conflicts of Interest:** The authors declare no conflict of interest.

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
