# Peer review of "Image Denoising Based on GAN with Optimization Algorithm"

_electronics, doi:10.3390/electronics11152445_

Round 1

Reviewer 1 Report

May be it can be compared with wavelet or fractal methods of image processing?

Author Response

Point 1: May be it can be compared with wavelet or fractal methods of image processing?

Response 1:We compared the denoising algorithm with "Image denoising based on undecimated discrete Transform". It is shown in blue font in P.1/line43, Tables 1 and 2.

Reviewer 2 Report

Suggestions: 
1-P.1/line 35, p.1/line 38, p.2/line 47, p.2/line 48-49, p.2/line 50-51, p.2/line 52, p. 2/line 54: Use the same convention used for "Generative Adversarial Network (GAN)-p.1/line 11" in order to spell out "BM3D", "TWSC", "GVC", "CSF", "DnCNN" , "SCGAN", "DIDN". 
2-P.4, lines 146/147: Generalize the equation already in the definition of line 146. 
3-P.7, line 242: Replace "CIFAT10" with "CIFAR10". 
4-P.7/Table 1, p.8/Table 2: Highlight in bold the best values ​​for each noise case. 
5-P.7, Figure 4: slightly increase the size of the figures.

Author Response

Point 1: P.1/line 35, p.1/line 38, p.2/line 47, p.2/line 48-49, p.2/line 50-51, p.2/line 52, p. 2/line 54: Use the same convention used for "Generative Adversarial Network (GAN)-p.1/line 11" in order to spell out "BM3D", "TWSC", "GVC", "CSF", "DnCNN" , "SCGAN", "DIDN".

Response 1:We Use the same convention used for "Generative Adversarial Network (GAN)” for "BM3D", "TWSC", "GVC", "CSF", "DnCNN", "SCGAN", "DID", etc. As shown in red font P.1/line39, P.2/ line 50, P.2/ line 51, P.2/ line 53, P.2/ line 55, P.2/ line 57, P.2/ line 59;

Point 2: P.4, lines 146/147: Generalize the equation already in the definition of line 146.

Response 2: This is a generalization equation.

Point 3: P.7, line 242: Replace "CIFAT10" with "CIFAR10"..

Response 3:We change "CIFAT10" to "CIFAR10" in P.7/line 255.

Point 4: P.7/Table 1, p.8/Table 2: Highlight in bold the best values ​​for each noise case.

Response 4:We highlight in bold the best values ​​for each noise case in P.8/Tables 1 and P.8/Table2.

Point 5: P.7, Figure 4: slightly increase the size of the figures.

Response 5:We change the size of the font in P.7,Figure 4.

Reviewer 3 Report

In this paper, the authors proposed an Optimization Algorithm to train and optimize the mean and variance of noise on each node of the generator. Then the robustness of the denoising network is improved during the backpropagation. Overall, the topic of this paper is convincing and the problem is hot.  

Generative Adversarial Network(GAN) has been used for many years, please give us more explanations about why the model is novel by consulting with the book https://www.deeplearningbook.org/ or “Neural Networks and Learning Machines” by Haykin, S.O...

Please edit the English language and check the grammatical mistakes. There are some online tools to do it.

Image denoising is very popular recently, could the authors make some comparisons with

Cai, Shuting, et al. "Image denoising via improved dictionary learning with global structure and local similarity preservations." Symmetry 10.5 (2018): 167.

Cai, Shuting, et al. "A new development of non-local image denoising using fixed-point iteration for non-convex â„“p sparse optimization." PloS one 13.12 (2018): e0208503.

Author Response

Point 1: Generative Adversarial Network(GAN) has been used for many years, please give us more explanations about why the model is novel by consulting with the book https://www.deeplearningbook.org/ or “Neural Networks and Learning Machines” by Haykin, S.O...

Response 1:We provide a more comprehensive explanation of Generative adversarial networks (GAN), shown in yellow at P.3/line 115.

Point 2: Please edit the English language and check the grammatical mistakes. There are some online tools to do it.

Image denoising is very popular recently, could the authors make some comparisons with

Cai, Shuting, et al. "Image denoising via improved dictionary learning with global structure and local similarity preservations." Symmetry 10.5 (2018): 167.

Cai, Shuting, et al. "A new development of non-local image denoising using fixed-point iteration for non-convex â„“p sparse optimization." PloS one 13.12 (2018): e0208503.

Response 2:We use some online tools check the grammatical mistakes

We make a comparison with "Image Denoising Via Improved Dictionary Learning with Global Structure and Local Similarity Preservations". As shown in yellow font in P.2/line67, Table 1 and Table 2;

We consider "A new development of non-local image denoising using fixed-point iteration for non-convex â„“ P sparse "Optimization" is referenced, as shown in the P2/line70 yellow font.
